# Zinc: A Necessary Ion for Mammalian Sperm Fertilization Competency

**DOI:** 10.3390/ijms19124097

**Published:** 2018-12-18

**Authors:** Karl Kerns, Michal Zigo, Peter Sutovsky

**Affiliations:** 1Division of Animal Sciences, University of Missouri, Columbia, MO 65211-5300, USA; kkerns@mail.missouri.edu (K.K.); zigom@missouri.edu (M.Z.); 2Department of Obstetrics, Gynecology and Women’s Health, University of Missouri, Columbia, MO 65211-5300, USA

**Keywords:** fertilization, sperm, capacitation, zinc, proteasome, fertility

## Abstract

The importance of zinc for male fertility only emerged recently, being propelled in part by consumer interest in nutritional supplements containing ionic trace minerals. Here, we review the properties, biological roles and cellular mechanisms that are relevant to zinc function in the male reproductive system, survey available peer-reviewed data on nutritional zinc supplementation for fertility improvement in livestock animals and infertility therapy in men, and discuss the recently discovered signaling pathways involving zinc in sperm maturation and fertilization. Emphasis is on the zinc-interacting sperm proteome and its involvement in the regulation of sperm structure and function, from spermatogenesis and epididymal sperm maturation to sperm interactions with the female reproductive tract, capacitation, fertilization, and embryo development. Merits of dietary zinc supplementation and zinc inclusion into semen processing media are considered with livestock artificial insemination (AI) and human assisted reproductive therapy (ART) in mind. Collectively, the currently available data underline the importance of zinc ions for male fertility, which could be harnessed to improve human reproductive health and reproductive efficiency in agriculturally important livestock species. Further research will advance the field of sperm and fertilization biology, provide new research tools, and ultimately optimize semen processing procedures for human infertility therapy and livestock AI.

## 1. Introduction—Encyclopedism of Biological Zinc

Zinc (Zn) is one of the most highly abundant elements on earth, an essential micronutrient to all things living, typically occurring as a divalent cation metal with moderate reactivity and reducing properties. Essential biological roles of zinc include signaling, enzymatic activities, regulation of normal growth and sexual maturation, digestion, homeostasis of central nervous system, and mitochondrial oxidative stress [1,2]. Conversely, zinc imbalance or altered zinc-signaling accompanies pathologies including but not limited to Alzheimer’s disease [3,4,5], blindness, cancer, digestive ailments, growth retardation, and inflammation [6]. While ancient Etruscans and Romans may have already recognized medicinal properties of zinc salts [7], its biological importance was only fully realized in the 19th century, and it entered the mainstream human medicine hundred years later, when the first studied were conducted on dwarfism, human zinc deficiency, and general importance of zinc as a growth factor [8].

Cells of all organisms, ranging from *E. coli* to mammals, tightly regulate free zinc ion (Zn^2+^) distribution, even though its toxicity is relatively low [9,10]. In humans, nearly 90% of Zn^2+^ is found in the muscle and bone [11]. Other organs containing significant concentrations of Zn^2+^ are the prostate, liver, gastrointestinal tract, kidney, skin, lung, brain, heart, and pancreas [12,13,14]. Homeostasis of Zn^2+^ is important for survival and fitness; thus, when Zn^2+^ is consumed in excess, it is important for the body to handle its surplus [15]. Upon ingestion and absorption through the small intestine, the redistribution of Zn^2+^ occurs via the serum, where Zn^2+^ is bound predominantly to albumin (major binding protein for up to 60% of Zn^2+^); the remaining Zn^2+^ is bound predominantly to 12 other proteins, including α_2_-macroglobulin, transferrin, ceruloplasmin, IgG, IgA, IgM, complement C4, haptoglobin, and prealbumin [16,17]. Serum Zn^2+^ accounts for only ~0.1% of bodily Zn^2+^ [18]. Further, there is no known specialized Zn^2+^ storage system in the body, and therefore only the daily intake of Zn^2+^ will ensure steady availability [17].

On the cellular level, 30–40% of Zn^2+^ localizes in the nucleus, while 50% is stored in the cytoplasm and the rest is associated with membranes [19]. There are two families of proteins that are responsible for the movement of Zn^2+^ through biological membranes, thus exercising sustained homeostatic control. These include zinc-importer (ZIP; Zrt-, Irt-like) family proteins that transport Zn^2+^ into the cytosol and the zinc transporter (ZnT) family proteins transporting Zn^2+^ out of the cytosol [20]. Completion of human genome sequencing identified 14 members of ZIP (designated ZIP1–14) and 10 members ZnT (designated ZnT1–10) [20] families. Few studies have inspected major tissues for the expression patterns of ZnTs in humans [21], and the expression of ZIPs during spermatogenesis is only known in mice [22]. Once Zn^2+^ enters a cell by ZnTs and ZIPs, it becomes sequestered within the endoplasmic reticulum, mitochondria, and Golgi, or other cell type-specific membrane bound vesicular structures, also called zincomsomes [23,24]. Cytosolic Zn^2+^ complexing with cytosolic proteins maintains the concentration of free cytosolic Zn^2+^ within range between picomoles and nanomoles, depending on the cell type [9,25,26,27]. Up to 20% of cytosolic Zn^2+^ is bound by the apoprotein thionein, to form metallothionein (MT). MTs are small ubiquitous proteins (6–7 kDa) that are rich in cysteine that can complex transition metal ions [28,29]. One molecule of MT can bind up to seven Zn^2+^, buff excess Zn^2+^, and supply such cation under Zn^2+^ deficiency states [30,31]. 

Limited information exists on the regulation of Zn^2+^ homeostasis in reproductive system. In female gametes, Zn^2+^ plays a gatekeeping role in regulating meiotic resumption [32,33,34]. A novel phenomenon of Zn^2+^ release from the mammalian oocyte at fertilization was recently reported [33,35]; inspiring some of the work on male gametes that will be discussed later. The importance of Zn^2+^ for male fertility only emerged recently, propelled in part by consumer interest in nutritional supplements containing ionic trace minerals. Here, we review properties, biological roles, and cellular mechanisms that are relevant to zinc function in the male reproductive system, survey available peer-reviewed data on nutritional zinc supplementation for fertility improvement in livestock animals and infertility therapy in men, and discuss recently discovered signaling pathways involving zinc in sperm maturation and fertilization.

## 2. Zinc-Interacting Sperm Proteins

In eel, zinc is necessary for the sustenance of germ cells and spermatogenesis progression, with *N*,*N*,*N*′,*N*′-tetrakis(2-pyridinylmethyl)-1,2-ethanediamine (TPEN)-induced Zn deficiency causing germ cell apoptosis [36]. Zinc transporters have been examined in rat, specifically MT I & II in spermatocytes [37] as well as tesmin, a testis-specific metallothionein-like protein [38]. Further, ZnT-7 in mouse testis may supply spermatogenesis-required zinc [39]. Zn^2+^ transporter ZIP9 serves as a membrane-associated receptor interacting with G-protein Gnα11 to mediate the non-classical testosterone signaling cascade in murine spermatogenic GC-2 cells [40]. Further, the spontaneous Ca^2+^ oscillations that were observed in spermatogenic cells seem to be modulated by Zn^2+^ [41]. Zinc ions begin to colonize spermatogenic cells during the final stages of spermatid differentiation when they are incorporated in the nucleus [42] and nascent outer dense fibers (ODF) [43,44]. Additional Zn^2+^ is incorporated into the nucleus at ejaculation [45]. Nuclear Zn^2+^ associates with protamines and forms zinc bridges, most likely through imidazole groups of histidine and thiols of cysteine [46], as proposed by Bjorndahl and Kvist, to stabilize the sperm chromatin structure [47,48]. These authors showed that a rapid sperm chromatin decondensation can be induced by Zn^2+^ chelation with 2,2′,2′′,2′′′-(Ethane-1,2-diyldinitrilo) tetraacetic acid (EDTA), causing the disruption of the protamine zinc bridges [49,50,51]. In the sperm flagellum, Zn^2+^ is bound to sulfhydryl groups of ODF protein cysteine groups, to protect the nascent flagellum from premature oxidation [52]. During epididymal transit, Zn^2+^ is selectively removed from the flagellum by a 160 kDa protein, enabling the oxidation of sulfhydryl groups and stiffening the ODF to support progressive motility [52]. High concentrations of Zn^2+^ have been found in the acrosome [53] and proteolytic conversion of proacrosin to acrosin is inhibited by Zn^2+^ [54,55], as it probably involves Zn-dependent metalloproteinases. Zinc ions also associate with sperm membranes, where they interact with lipoproteins and membrane-bound metalloproteins in which they react with sulfhydryl groups of cysteine and therefore fulfill a membrane stabilizing function [56,57]. The active removal of Zn^2+^ is therefore a prerequisite for the completion of sperm capacitation [53], a complex structural and molecular remodeling event that endows spermatozoa within the female reproductive tract with the ability to fertilize. High concentrations of Zn^2+^ (100 µM) reduce sperm motility in a reversible manner [58]. Initiation of motility following ejaculation [59] and the increased motility of the capacitation-induced sperm hyperactivation are both dependent upon intracellular alkalinization [60].

Additional Zn^2+^ becomes incorporated into spermatozoa during ejaculation [21,45] where it is believed to have protective function in terms of sperm chromatin decondensation [47,48], sperm motility and metabolic inhibition [58,61], membrane stabilization [56], and antioxidant activity [62,63]. As Zn^2+^ becomes incorporated into spermatozoa upon mixing with seminal fluid, there are also seminal fluid Zn-interacting proteins competitively binding free Zn^2+^. In humans, a bulk of seminal fluid Zn^2+^ is bound to high and low molecular weight ligands derived from prostatic and vesicular secretions [64,65,66,67]. Among them, semenogelins participate in the formation of coagulum, to prevent the retrograde flow of semen deposited in the female tract. Prostasomes, small, exosome-like lipoprotein vesicles are the main zinc-binding partners in human seminal fluid [68,69]. Zinc-binding proteins have also been found in seminal fluid of boar [70] and dog [71,72], and are designated as ZnBP1–6.

## 3. Zinc-Containing Sperm Proteins 

Zinc-containing proteins, commonly known as metalloproteins, are capable of binding one or more Zn^2+^, usually as a requirement for their biological activity. Human genome sequencing and combined proteomic approaches independently identified 1684 proteins in the human proteome as zinc-containing proteins [73]. Metalloproteins can be further divided into three groups, i.e., (i) metalloenzymes, (ii) metallothioneins, and (iii) gene regulatory proteins [19,74]. Metallothioneins have been discussed in the previous section. Gene regulatory proteins are nucleoproteins that are directly involved with the replication and transcription of DNA. Such DNA binding proteins can be further categorized into three structurally distinct groups, containing: (i) zinc fingers, (ii) zinc clusters, or (iii) zinc twists [75]. Spermatozoa may have limited use for gene regulatory proteins since they are transcriptionally silent; however, these proteins, as, for instance, protamine P2 (discussed earlier) are used heavily for DNA condensation, packaging, and transcriptional suppression [76]. The majority of this section will therefore be dedicated to zinc-containing metalloenzymes, which play a vital role in sperm function.

More than 300 enzymes have been identified that require Zn^2+^ for their function [19], representing more than 50 different enzyme types. Zn^2+^ is the only metal that is encountered in all six classes of enzymes, (i.e., oxidoreductases, transferases, hydrolases, lyases, isomerases, and ligases). This can be attributed to two properties of Zn^2+^: (i) relatively low toxicity when compared to other transition metals [77] and (ii) stable association and coordination flexibility with macromolecules [78]. Zn^2+^ fulfills three functions in the Zn-enzymes: (i) catalytic, (ii) co-active (co-catalytic), and (iii) structural [79]. Catalytic Zn^2+^ takes part directly in enzyme catalysis. Co-active Zn^2+^ enhances or diminishes catalytic function in conjunction with catalytic Zn^2+^, but it is not indispensable for catalytic function [79]. Structural Zn^2+^ is required for stabilization of the quaternary structure of oligomeric enzymes.

Matrix metalloproteinases (MMPs) belong to a family of zinc-dependent endopeptidases, which are involved in the degradation of extracellular matrix proteins. Since the first discovery of MMPs in the early 1960s, MMPs have grown in number and at least 28 species have been identified to this date; for subtype categorization, distribution, and substrate specificities, see review by Cui et al. [80]. Structurally, a typical MMP contains a propeptide, a catalytic metalloproteinase domain, a linker peptide (hinge region), and a hemopexin domain [80]. The catalytic domain contains two Zn^2+^ (catalytic and structural) and up to three calcium ions (Ca^2+^), which stabilize the structure. The cysteine rich region in propeptide chelates the catalytic Zn^2+^, keeping MMPs in an inactive zymogen form [81]. MMP2 and MMP9, also referred to as Gelatinase-A and Gelatinase-B, were described in human seminal fluid [82,83] and canine epididymal fluid and seminal fluid [84,85]. Furthermore, MMP2 was found to be localized in acrosomal and tail region of normal morphological ejaculated human and canine spermatozoa, while MMP9 was localized in the tail region [85,86]. High levels of MMP2 are associated with high (70%) motility and significantly elevated levels of MMP9 are observed in semen samples with low sperm count [85]. Ferrer at al. [87] demonstrated that MMP2 together with acrosin were confined to the inner acrosomal membrane of epididymal bull sperm and thus introducing the possibility of their cooperation in enzymatic digestion of the oocyte zona pellucida (ZP) during penetration. The regulation of said MMPs by zinc ion fluxes associated with sperm capacitation is currently under investigation. Kratz et al., [88] demonstrated that the levels of seminal MMP2 and MMP9 are correlated with oxidative stress in men, making this a potential diagnostic tool for semen quality/male infertility. Finally, Atabakhsh et al. [89] noticed a positive correlation between seminal fluid MMP2 activity and sperm count, as well as fertilization and embryo quality in couples undergoing assisted reproductive therapy (ART) by intracytoplasmic sperm injection (ICSI), offering a potential predictor of ICSI outcome.

Superoxide dismutases (SOD) are metalloenzymes that are responsible for dismutating the superoxide anion (O_2_^−^), to hydrogen peroxide (H_2_O_2_), and oxygen (O_2_) [90]. Three isoforms have been reported in mammals: (i) the cytosolic dimeric Cu/Zn-SOD (SOD1), (ii) the mitochondrial matrix Mn-SOD (SOD2), and (iii) the secretory tetrameric extracellular SOD (EC-SOD/SOD3) [91]. It was shown earlier that both seminal fluid and spermatozoa contain SOD activity [92,93,94,95,96,97,98,99], of which 75% was attributed to SOD1, which is also the main SOD isoform in spermatozoa SOD. The SOD activity in spermatozoa is several-fold higher than SOD activity levels that were previously measured in more than 50 different human somatic cell types [98]. O’Flaherty et al. [94] suggested an important role of superoxide anion in sperm hyperactivation and capacitation; therefore, an adequate balance between superoxide radical generation and dismutation is vital for proper function of spermatozoa, as implicated by Sikka [100].

Another significant group of Zn^2+^ containing proteins of spermatozoa are sorbitol dehydrogenases that convert sorbitol to fructose, and endow spermatozoa with and have been correlated to motility [101]. Lactate dehydrogenase isoenzyme (LDH-X, LDH-C_4_) also has been reported to have relationship with sperm motility [102,103,104]. It was shown at least in mice that the inhibition of LDH-C_4_ blocked sperm capacitation [105]. We previously reported the presence of a ring finger ubiquitin ligase homologous to UBR7 in round spermatids and spermatozoa [106], and implicated this zinc finger containing enzyme in spermiogenesis and possibly in the proteolytic degradation of the ZP at fertilization [107]. Angiotensin converting enzyme (ACE), yet another important Zn^2+^ containing protein, has been reported in testis, epididymis, and spermatozoa of stallion, boar, and man [108,109,110,111,112,113,114]. Several roles in reproduction have been proposed for ACE, including spermatogenesis [115], sperm capacitation [116,117], and sperm-ZP binding [118]. Alkaline phosphatase (ALP), a homodimeric enzyme containing two Zn^2+^ and one Mg^2+^ is present in mammalian seminal fluid [119,120] and spermatozoa [121]. The precise role of ALP in reproduction remains to be discovered, though it may serve as a decapacitating factor [122]. Additional Zn^2+^ containing proteins that are found in the spermatozoa include fructose-bisphosphate aldolases [123], of which class-II possesses Zn^2+^ [124], and alcohol dehydrogenase present in human testis and spermatozoa [125,126]. The ADAM (A Disintegrin and Metalloproteinase) protein family plays a role in multiple events leading up to fertilization, including gamete migration and sperm reservoir interactions, sperm-oocyte ZP binding, and sperm-oocyte plasma membrane adhesion and fusion (see review [127]). Noteworthy, proteins in this family possess metalloproteinase domains with Zn^2+^ [128]. The metalloproteinase domain, however, is cleaved during epididymal transit and only the disintegrin domain remains in mature spermatozoon [127]. Altogether, it is likely that the zinc-interacting proteome plays varied and often essential roles in the regulation of sperm homeostasis and fertilizing ability. Rather than a complete list of zinc-containing proteins, we focused on proteins that are well characterized. We are aware that there are many zinc-containing proteins to be characterized in spermatozoa.

## 4. Zinc as a Regulator of Sperm Capacitation and Fertilization

Zinc ions play a vital role in sperm capacitation, regulating key events that are responsible for fertilization competency (summarized in Figure 1a). As much as the Ca^2+^ influx was understood as key for capacitation, today it is understood that the Zn^2+^ efflux is the gatekeeper to this important Ca^2+^ influx [129,130,131,132]. In the following discussion of sperm capacitation, it is important to note the contrasting definitions of sperm capacitation (physiological vs. biochemical) [133] and we will discuss it strictly from the earlier in its original definition (the acquisition of the capacity to fertilize [134]). Prior to the discovery of the sperm capacitation state-reflecting Zn signatures in higher order mammals (boar, bull, and human) [135], there was a noticeable paucity of pivotal discoveries in sperm capacitation translatable from rodent models to humans [136]. Much of this was criticized as a lack of in vivo or minimal inclusion of an in vitro female component in sperm capacitation studies; however, a critical review of literature suggests that this could be due to subtle but vivid differences in the study models and/or experimental design. This includes species differences in attaining intracellular alkalinization [137], thus regulating Ca^2+^ entry (solely the Na^+^-dependent Cl^−^/HCO_3_^−^ exchanger [138] and possibly the sperm-specific Na^+^/H^+^ exchanger sNHE [139] in murine; hydrogen voltage-gated channel, Hv1 expressed by *HVCN1* in humans [140]), as well as a result of using epididymal spermatozoa (as opposed to ejaculated). Both of these factors have a notable impact on the Zn signature and result in studies that do not mimic the physiology of ejaculated human semen.

There is a moderate negative correlation between flagellar Zn^2+^ content, and sperm global and progressive motility in humans [141]. Chelation of sperm Zn^2+^ by (2*R*,3*S*)-2,3-Bis(sulfanyl)butanedioic acid (DMSA), 2,3-dimercaptopropane-1-sulfonate (DMPS), or dl-penicillamine leads to increased average straight line velocity and progressive sperm motility while decreasing the percentage of nonlinear motile spermatozoa [142]. Though discovered before the importance of Hv1 in sperm motility activation and capacitation surfaced, previous authors believed this Zn^2+^ removal to be solely associated with the stiffening of the ODF. Voltage-gated proton channel, Hv1 localizes to the sperm flagellum and it is responsible for sperm cytoplasmic alkalinization through transmembrane proton extrusion [61]. Hv1 is asymmetrically positioned, likely providing differing alkalized microenvironments and gradients in relationship to the symmetrically positioned CatSper channels, thereby being responsible for asymmetrical flagellar bending during hyperactivation [143]. 

The sperm Zn signature is a collective term for four distinct Zn^2+^ localization patterns that are indicative of the sperm capacitation state [135]. These zinc ion fluxes are associated with key events in the acquisition of fertilization competency, indicating non-capacitated state, hyperactivation, acrosomal modifications, and acrosomal exocytosis (summarized in Figure 1b). These distinct signatures minimally distinguish the sequential sperm capacitation subpopulations, or to the extent that Zn^2+^ establishes these sequential subpopulations, and thereby is the previously unknown regulatory time clock of sperm capacitation. The decrease in Zn^2+^ concentration from the ejaculation/deposition site to the site of fertilization could remove sperm Zn^2+^ simply via concentration gradient differences and the filtering out of seminal fluid, thereby promoting sperm capacitation. Further, it is well understood that the ubiquitin-dependent protease holoenzyme, the 26S proteasome regulates sperm capacitation (review [144]), as well as the Zn^2+^ flux in boar spermatozoa [135], and participates in sperm acrosomal exocytosis induced by binding to the egg coat in sea urchin [145], bull [146], and human [147] spermatozoa. Besides acrosomal exocytosis, the sperm-borne 26S proteasome has been implicated in egg coat penetration (ascidian, sea urchin [145], and boar [107]). The relationship between 26S proteasome activity and Zn^2+^ is unclear, though the proteasomal regulatory subunit PSMD14/Rpn11 contains a metalloprotease-like Zn^2+^ site [148]. Additionally, Zn^2+^ has been implicated in regulating proteasome-dependent proteolysis in HeLa cells [149]. Contrarily to the high seminal fluid Zn^2+^ concentrations (2 mM) inhibiting Hv1, lower concentrations (20 and 50 µM) have been implicated in promoting acrosomal exocytosis in sea urchin [150] and bovine [151] spermatozoa during in vitro capacitation. It is believed that Zn^2+^ interacts with Zn-sensing receptor (ZnR) GPR39 of the G-protein-coupled receptor (GPCR) family that is found in the sperm acrosome. Such interaction stimulates acrosomal exocytosis through epidermal growth factor receptor (EGFR) transactivation and the phosphorylation of phosphoinositide 3-kinase (PI3K) causing acrosomal Ca^2+^ mobilization [151]. This implicates a multifaceted role of Zn^2+^ in sperm capacitation and therefore more research will be needed to fully comprehend these contrasting pathways (inhibiting vs. inducing acrosomal exocytosis). 

Successful embryo development in mammals depends upon efficient anti-polyspermy defense, preventing the entry of more than one spermatozoon in the oocyte cytoplasm at fertilization, and thus alleviating an embryo-lethal polyploidy. While membrane depolarization and cortical granule exocytosis are regarded as the main barriers to polyspermy, a sperm-induced Zn^2+^ release from the oocyte cortex, nicknamed the Zn^2+^ spark, has recently been discovered in mammals [33,35]. Besides the oocyte Zn^2+^ spark [33], there is also a physiochemical ZP hardening and a 300% Zn^2+^ increase in the ZP matrix observed when the fertilized oocyte zona becomes refractory to sperm binding in the mouse [35]. Such a proposed new anti-polyspermy defense mechanism is plausible, although the exact mechanism was not known until recently. We now know that Zn^2+^ is chemorepulsive, possibly overriding the chemoattraction of oocyte-secreted progesterone in capacitated human, mouse, and rabbit spermatozoa [152]. In light of the sperm Zn signature, a polyspermy defense mechanism of newly fertilized oocytes, termed the zinc shield [135], could in fact de-capacitate spermatozoa already bound to the zona or present in the perivitelline space at the time of fertilization. It is likely such hijacking sperm Zn-signaling and de-capacitation complements the blockage of fertilization through traditional anti-polyspermy mechanisms. Zn^2+^ has been shown to inhibit fertilization when added to bovine in vitro fertilization (IVF) media [153]. In further support of this mechanism, Zn^2+^ regulates the activity of the proposed sperm-borne ZP lysins, the proteinases [87] implicated in ZP penetration, including acrosin [54,55], 26S proteasome [135,144], and MMP2 (brain) [154], therefore playing a regulatory role in sperm-ZP penetration. Additionally, inhibitors of zinc-dependent metalloproteases hinder sperm passage through the cumulus oophorus during porcine IVF [155].

## 5. Effect of Zinc Supplementation on Male Fertility

Reduced seminal fluid Zn^2+^ has been reported in cases of male infertility that are associated with accidental Chernobyl radiation in Ukraine [156], signifying a possible relationship between Zn^2+^ and male fertility. Indeed, seminal fluid Zn^2+^ concentration is positively correlated with sperm count and normal sperm morphology [157]. Consequently, Zn^2+^ supplementation improves sexual dysfunction in rats [158] and uremic men [159], which is likely due to the ability of Zn^2+^ to increase serum testosterone levels [160]. The negative effect of fatiguing bicycle exercise on thyroid hormone and testosterone levels in sedentary males is likewise prevented with Zn^2+^ supplementation [161]. Oral Zn^2+^ supplementation results in increased sperm counts in ram [162] and humans (combined with inclusion of folate; review) [163]. Zinc supplementation also restores superoxide scavenging antioxidant capacity in asthenospermic men [164]. Dietary Zn^2+^ intake and action on intraprostatic Zn^2+^ levels remain unknown; however, if such supplementation increases prostatic levels, it could perceivably increase the percent of non-capacitated signature 1 spermatozoa at the time of ejaculation. Such would be beneficial for inhibiting Hv1, warding off premature sperm capacitation. Goat dietary Zn^2+^ supplementation increases sperm plasma membrane and acrosome integrity, and percent of viable spermatozoa, also increasing seminal fluid SOD, catalase, and glutathione peroxidase activities [63]. Additionally, Cu^2+^ and Zn^2+^ dietary co-supplementation to bucks of the Osmanabadi goat breed allowed for them to reach puberty 28–35 days earlier [165]. While Zn^2+^ supplementation has a positive influence on multiple male reproductive measures and can serve as male infertility therapy under certain circumstances, one study suggests that supplementation over 100 mg/day is associated with a 2.29 relative risk of advanced prostate cancer [166]. The exact mechanism is unclear, as Zn^2+^ is not generally considered a carcinogen [167]; yet, increased risk is not surprising as zinc becomes cytotoxic at such concentrations and warrants dosage caution.

Few studies have assessed the addition of Zn^2+^ in media/semen extenders in agriculturally important livestock species propagated by artificial insemination (AI). A highly desirable property of such media is to reduce reactive oxygen species (ROS) that effect sperm function by the oxidation of lipids, proteins, and DNA (review [168]). Notably, Zn^2+^ supplementation would serve as an antioxidant by scavenging excessive superoxide anions [169], though Zn^2+^ homoeostasis is important, and too much Zn^2+^ can act as a prooxidant and cause mitochondrial oxidative stress (review [2]). Some studies have investigated Zn^2+^ supplementation, however they did so with the inclusion of d-aspartate and coenzyme Q10, without distinguishing which compound positively reduced lipid peroxidation and DNA fragmentation [170] and improved embryo development [171]. Spermatozoa have the capacity for Zn^2+^ loading [135], to the extent of restoring their pre-capacitation Zn signature, and it seems reasonable that such would reduce premature, pathological sperm capacitation. 

## 6. Spermatotoxicity and Reprotoxicity of High Zinc Contamination and Zinc Deficiency

At high soil levels, Zn^2+^ is reprotoxic to the terrestrial worm *Enchutraeus crypticus* [172]. Few studies exist that observe Zn^2+^ reprotoxicity [173]. ZnCl_2_ dietary supplementation to both male and female rats at 30 mg/kg/day but not 15 or 7.5 mg/kg/day showed significant reduction in fertility, offspring viability, and body weight of F1 pups; however, it had no effect on litter size (male reprotoxicity alone was not observed) [174]. Nanosized ZnO toxicity induces sea urchin sperm DNA damage, but it does not reduce fertility [175]. A moderate negative correlation (*r* = −0.426) has been found between total flagellar Zn^2+^ content and the percentage of morphologically normal spermatozoa in men [141]. Morphologically abnormal spermatozoa actually contained high amounts of Zn^2+^ (as reported by fluorescent Zn-probe) [135]. Whether such is caused by Zn^2+^ toxicity or simply a product of defective spermatozoa failing to regulate their ion fluxes is unknown. Dietary Zn^2+^ deficiency studies in rat have shown reduced relative weight of the testes, epididymides, and seminal vesicles (organ by percentage of body weight) and reduced prostate weight (by organ weight alone). In the same study, the length of the flagellum was reduced on average by 24% in Zn^2+^ deficient rats, accompanied by a substantial increase in morphologically abnormal spermatozoa (especially abnormal heads, predominantly headless). Some of these defects could be due to abnormal testicular function from modulated fatty acid composition, interrupting essential fatty acid metabolism, where ω-6 polyunsaturated fatty acids were highly enriched [176]. Zn^2+^ deficiency is known to trigger autophagy in yeast [177], and elevated autophagy rate during spermatogenesis could decrease sperm count during Zn^2+^ deficiency. In the absence of fertilization, sperm capacitation is a terminal event leading down a rapid path of apoptosis [178], possibly from the overproduction of ROS [179] in an environment with reduced Zn^2+^. No studies have been performed to observe whether micromolar levels of Zn^2+^ under capacitation-inducing conditions can prolong sperm lifespan during a fertilization competent state (as opposed to millimolar levels which inhibit sperm Zn^2+^ flux [135]). If such could be achieved, fertilization would be possible with fewer spermatozoa and especially useful for artificial insemination in livestock as well as human intrauterine insemination (IUI) in place of costly IVF treatments of couples with an oligospermic male partner.

## 7. Conclusions and Perspectives 

Through a variety of pathways, Zn^2+^ plays a gatekeeping role in male gametes just as it does in those of the female. Prostatic seminal fluid with the highest concentration of Zn^2+^ found in any bodily fluids plays a crucial role in fending off premature sperm capacitation and it provides antioxidant activity, while lower concentrations of Zn^2+^ may be a prerequisite for successful acrosomal exocytosis. A list of reviewed Zn-containing and interacting proteins that are involved from spermatogenesis through the final preparatory steps of fertilization is summarized in Table 1. To fully understand the biological role of Zn^2+^ in male fertility, further research needs to be pursued, especially to fully disclose the Zn-interacting sperm proteome and its place in various cellular pathways controlling male reproductive function. Additionally, dietary and semen media Zn^2+^ supplementation has been found to be beneficial for male fertility. Collectively, the currently available data already hint at the importance of zinc ions for male fertility, which could be harnessed to improve the reproductive performance of livestock and increase the success rate of human assisted reproductive therapy. Further research will advance the field of sperm and fertilization biology, provide new research tools, and ultimately optimize the semen processing procedures for human infertility therapy and livestock artificial insemination.

## Figures and Tables

**Figure 1 ijms-19-04097-f001:**
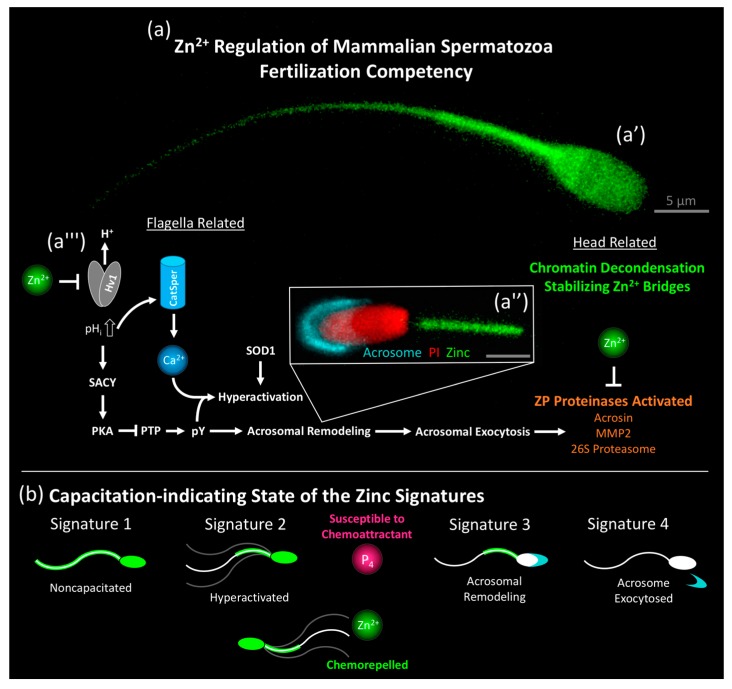
Summary of Zinc (Zn) signatures and free zinc ion (Zn^2+^) regulation of the fertilization competency of mammalian spermatozoa. (**a**) Super-resolution images of the non-capacitated boar sperm Zn signature 1 (**a’**) and acrosome-remodeled sperm Zn signature 3 (**a''**) acquired by the Leica TCP SP8 stimulated emission depletion (STED) microscope (free zinc ions in green, outer acrosomal membrane in cyan, remodeled sperm head plasma membrane in red; scale bars in gray: 5 μm). (**a'''**) High Zn^2+^ concentration (2 mM) negatively regulates proton channel Hv1, responsible for the rise of intracellular pH, facilitating: (1) Ca^2+^ entry via CatSper and (2) protein tyrosine phosphorylation (pY), triggered by activation of soluble sperm adenylyl cyclase (SACY), increasing intracellular cAMP, activating protein kinase A (PKA) and phosphorylating protein tyrosine phosphatases (PTP) to an inactive state. For general capacitation pathway, review see Kerns et al., [144]). Following acrosome remodeling and exocytosis, zona pellucida (ZP) proteinases (acrosin, MMP2, and the 26S proteasome) implicated in endowing the spermatozoon with the ability to penetrate the ZP are activated. Zn^2+^, abundantly present in the fertilizing sperm triggered oocyte zinc shield, negatively regulates proteinase activities of spermatozoa bound to the zona or present in the perivitelline space, de-capacitating spermatozoa and serving as a newly proposed anti-polyspermy defense mechanism. (**b**) Capacitation-indicating state of the zinc signatures. Signature 1 spermatozoa are in a non-capacitated state. Signature 2 spermatozoa display hyperactivated motility. Only capacitating spermatozoa susceptible to progesterone (P_4_) chemoattraction exhibit chemorepulsion by Zn^2+^. Signature 3 spermatozoa exhibit acrosome remodeling while acrosomal exocytosis reportedly occurs in signature 4.

**Table 1 ijms-19-04097-t001:** General summary of testicular/sperm Zn-containing and interacting proteins reviewed, in order of discussion.

Protein	Localization ^1^	Function ^2^	Reference
Metallothionein I & II	Spermatocytes (rat)	Zinc ion binding	[37]
Tesmin	Spermatocytes (rat)	Developmental protein	[37]
Zinc transporter ZIP9	Spermatogenic GC-2 cells (rat)	Zinc transmembrane transporter activity	[38]
Protamine-2	Sperm nucleus (man)	Developmental protein, DNA-binding	[42,47,48]
Acrosin	Acrosome (man, bull)	Hydrolase, protease, serine protease	[54,55]
Semenoglins	Prostatic and vesicular fluids (man)	-	[64,65,66,67]
ZnBP1–6	Seminal fluid (boar, dog)	-	[70,71,72]
Matrix metalloproteinase-2	Seminal fluid, inner acrosomal membrane, flagellum (man, bull)	Hydrolase, metalloprotease, protease	[87]
Matrix metalloproteinase-9	Seminal fluid and flagellum (man, bull)	Hydrolase, metalloprotease, protease	[87]
Superoxide dismutase 1, 2, 3	Mitochondria and seminal fluids (man)	Antioxidant, oxidoreductase	[91,92,93,94,95,96,97,98,99]
l-lactate dehydrogenase	Seminal fluids (man)	Oxidoreductase	[102,103,104]
Putative E3 ubiquitin-protein ligase UBR7	Sperm inner acrosomal membrane (boar)	Transferase	[105]
Angiotensin converting enzyme	Testis, epididymis, and spermatozoa (man, boar, stallion)	Carboxypeptidase, hydrolase, metalloprotease, protease	[107]
Alkaline phosphatase, germ cell type	Seminal fluid and sperm plasma membrane (man, boar)	Alkaline phosphatase	[119,120,121]
Putative class-II fructose-bisphosphate aldolase	Spermatozoa (bull)	Lyase	[123]
Alcohol dehydrogenase class-3	Testis, spermatozoa (man)	Oxidoreductase	[125,126]
ADAMs	Spermatozoa	Integrin binding, metalloendopeptidase	Review [127]
Voltage-gated hydrogen channel 1	Flagellum (man)	Ion channel	[140]
G-protein coupled receptor	Acrosome (bull)	G-protein coupled receptor activity	[151]
26S proteasome non-ATPase regulatory subunit 14	Inner acrosomal membrane	Hydrolase, metalloprotease, protease	[180]

**^1^** Including species reported. ^2^ Known or predicted function.

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
