# Peer review of "Zinc: A Necessary Ion for Mammalian Sperm Fertilization Competency"

_ijms, 2018, doi:10.3390/ijms19124097_

Reviewer 1 Report

This is a really nice review, broadening the field of reproduction physiology.

There are just few suggestions to the authors to consider to improve the presented paper.

Chapter 1.:

It would be relevant to start targeting the reproductive field in certain parts, such as explore briefly on knowledge of mitochondrial oxidative stress, testicular cancer, infertility therapy in men…..

Chapter 2.:

It is rather short, and could be extended in many parts with more detailed knowledge, esp. second paragraph.

Chapter 3.:

ADAM would be worth explaining in slightly more detail its role not only stating “some role….” 

A complete list (table) of zinc-containing proteins relevant and they function, predicted function or they characterization known at least from somatic cells would be useful, even though they are not fully characterized in sperm. This would be good to present a current knowledge on the topic. 

Chapter 5.:

More structures comparison between men and farm / laboratory animals would be interesting, if data available. Also, the last paragraph touching prostate cancer is not very informative and deserves addressing.

Author Response

Reviewer 1

This is a really nice review, broadening the field of reproduction physiology. There are just few suggestions to the authors to consider to improve the presented paper.

Chapter 1.: It would be relevant to start targeting the reproductive field in certain parts, such as explore briefly on knowledge of mitochondrial oxidative stress, testicular cancer, infertility therapy in men.

RESPONSE: Thank you for your suggestions. We intended the introduction to discuss the general biological role of zinc. With your suggestion, we have included the prooxidant property of zinc briefly in our discussion near the discussion of its antioxidant properties in section 5. We cover Zn2+ supplementation for male infertility therapy under section 5. Authors found no significant information regarding the role of zinc in testicular cancer.

Chapter 2.: It is rather short, and could be extended in many parts with more detailed knowledge, esp. second paragraph.

RESPONSE: We have expanded the first part of section 2 (related to maintenance of germ cells). While we would prefer to expand on the second paragraph, we are already exceeding the IJMS suggested manuscript length (4000 words).

Chapter 3.: ADAM would be worth explaining in slightly more detail its role not only stating “some role….” 

A complete list (table) of zinc-containing proteins relevant and they function, predicted function or they characterization known at least from somatic cells would be useful, even though they are not fully characterized in sperm. This would be good to present a current knowledge on the topic. 

RESPONSE: We have added a complete summary table of the zinc-containing and interacting proteins reviewed at the manuscript conclusion and explained ADAM in more detail.

Chapter 5.: More structures comparison between men and farm / laboratory animals would be interesting, if data available. Also, the last paragraph touching prostate cancer is not very informative and deserves addressing.

RESPONSE: We edited the prostate cancer sentence to make it clearer and moved it to the end of the first paragraph in section 5. We made sure to distinguish between men, and farm and laboratory animals, and re-reviewed the literature for further comparison. Data is limited, however, we have added one more citation. The main comparison regarding differences involving zinc between species can be found in section 4 (HVCN1).

Reviewer 2 Report

Kerns et al. expose in this work a significant update about the biological role of Zn in mammalian sperm function. The review is well written (extensive bibliographic revision) and deals with the main fields of sperm physiology. Nevertheless, I recommend the authors to include some references (see below) regarding the importance of Zn on spermatogenesis, sperm morphology, and male fertility (among others).

I miss some information about the role of Zn for the maintenance of germ cells and the progression of spermatogenesis (e.g.: Yamaguchi et al., Proceedings of the National Academy of Sciences, 2009; Shihan et al., Cellular Signalling, 2015; López-González et al., Journal of Cellular Physiology, 2016).

In line 278 the authors mention that low amount on Zn in seminal fluid has been found in men with fertility disorders associated to the Chernobyl accident highlighting the possible implications of Zn in male infertility. In fact, the relationship of Zn levels in seminal plasma with sperm quality from fertile and infertile men was reported ten years ago (Colagar et al., Nutrition Research, 2009). This reference needs to be included in this section.

It has been reported (Merrells et al., British Journal of Nutrition, 2009) that a Zn deficient diet significantly reduces the length of sperm flagellum, the percentage of normal sperm, and the mass of testes, epididymis, prostate and seminal vesicles in rats. In the same work, authors also demonstrated how a severe Zn deficiency modulates testis fatty acid composition. The results from this study should be mentioned within the 5 and 6 sections.  

In line 205, please replace average velocity straight line by straight line velocity (VSL). 

Author Response

Reviewer 2

Kerns et al. expose in this work a significant update about the biological role of Zn in mammalian sperm function. The review is well written (extensive bibliographic revision) and deals with the main fields of sperm physiology. Nevertheless, I recommend the authors to include some references (see below) regarding the importance of Zn on spermatogenesis, sperm morphology, and male fertility (among others).

I miss some information about the role of Zn for the maintenance of germ cells and the progression of spermatogenesis (e.g.: Yamaguchi et al., Proceedings of the National Academy of Sciences, 2009; Shihan et al., Cellular Signalling, 2015; López-González et al., Journal of Cellular Physiology, 2016).

In line 278 the authors mention that low amount on Zn in seminal fluid has been found in men with fertility disorders associated to the Chernobyl accident highlighting the possible implications of Zn in male infertility. In fact, the relationship of Zn levels in seminal plasma with sperm quality from fertile and infertile men was reported ten years ago (Colagar et al., Nutrition Research, 2009). This reference needs to be included in this section.

In line 205, please replace average velocity straight line by straight line velocity (VSL). 

RESPONSE: Thank you for your kind comments. We have incorporated all of the above comments and references in the revised manuscript.

It has been reported (Merrells et al., British Journal of Nutrition, 2009) that a Zn deficient diet significantly reduces the length of sperm flagellum, the percentage of normal sperm, and the mass of testes, epididymis, prostate and seminal vesicles in rats. In the same work, authors also demonstrated how a severe Zn deficiency modulates testis fatty acid composition. The results from this study should be mentioned within the 5 and 6 sections.  

RESPONSE: We have incorporated the above comment and reference in section 6 of the revised manuscript.